# Dirichlet Continual Learning: Tackling Catastrophic Forgetting in NLP

Min Zeng[1]     Haiqin Yang[2]     Wei Xue[1]     Qifeng Liu[1]     Yike Guo[*1]

[1]Hong Kong University of Science and Technology
[2]International Digital Economy Academy (IDEA)

## Abstract

Catastrophic forgetting poses a significant challenge in continual learning (CL). In the context of Natural Language Processing, generative-based rehearsal CL methods have made progress in avoiding expensive retraining. However, generating pseudo samples that accurately capture the task-specific distribution remains a daunting task. In this paper, we propose Dirichlet Continual Learning (DCL), a novel generative-based rehearsal strategy designed specifically for CL. Different from the conventional use of Gaussian latent variable in Conditional Variational Autoencoder, DCL employs the flexibility of the Dirichlet distribution to model the latent variable. This allows DCL to effectively capture sentence-level features from previous tasks and guide the generation of pseudo samples. Additionally, we introduce Jensen-Shannon Knowledge Distillation, a robust logit-based knowledge distillation method that enhances knowledge transfer during pseudo-sample generation. Our extensive experiments show that DCL outperforms state-of-the-art methods in two typical tasks of task-oriented dialogue systems, demonstrating its efficacy.

## 1 INTRODUCTION

Continual learning (CL) is a significant learning paradigm that aims to emulate human capacity for continuous learning and knowledge accumulation, while also ensuring that previously learned knowledge is retained and effectively transferred to facilitate the learning of new tasks [Parisi et al., 2019a]. However, in practice, models often encounter the notorious issue of *catastrophic forgetting* (CF), which refers to the phenomenon of models forgetting previously

---

[*]Corresponding Author

learned tasks when learning new tasks [McCloskey and Cohen, 1989, Parisi et al., 2019b]. This challenge is particularly prominent in the context of Natural Language Processing (NLP), where the complexity and diversity of language pose additional difficulties for CL [Ke and Liu, 2022b, Mehta et al., 2023].

To address the CF issue, various approaches have been proposed: (1) *Regularization* methods aim to minimize updates to the important parameters of previous tasks, thus preserving their performance [Kirkpatrick et al., 2017, Zenke et al., 2017, Aljundi et al., 2018]. However, the accumulation of regularizers may overly constrain network parameters, hindering the learning of new tasks. (2) *Architectural* approaches modify the network structure to enhance the extraction of task-specific features [Serra et al., 2018, Ke et al., 2021, Madotto et al., 2021, Zhang et al., 2022]. However, their task-focused approach may overlook effective knowledge transfer across tasks. (3) *Rehearsal* strategies involve replaying samples from previous tasks during training with the current task dataset [Lopez-Paz and Ranzato, 2017, Sun et al., 2019, Rolnick et al., 2019, Chuang et al., 2020, Mi et al., 2020a,b, Zhao et al., 2022a]. Rehearsal methods can be categorized into store-based rehearsal and generative-based rehearsal. It is worth noting that rehearsal methods have shown promise in mitigating forgetting in CL. However, store-based rehearsal may result in inefficiencies and increased memory demands, while generative-based rehearsal emerges as a more effective alternative. This approach facilitates efficient memory utilization and knowledge retention across sequential learning scenarios.

Generative-based rehearsal methods have emerged as promising approaches for mitigating the need for extensive retraining [Parisi et al., 2019b, Ke and Liu, 2022a]. These methods aim to generate pseudo samples that closely mimic the task-specific distribution, enabling the model to retain knowledge from previous tasks. However, generating such samples accurately remains a challenging task. The success of generative replay hinges on the production of high-quality pseudo samples that effectively approximate the real data

distribution of prior tasks. Higher-quality pseudo samples inherently contribute to better preservation of learned tasks, thereby minimizing forgetting in CL. However, previous studies [Sun et al., 2019, Chuang et al., 2020, Zhao et al., 2022a] have often demonstrated limited diversity, fluency, or poor alignment with the designated task when generating pseudo samples for each observed task.

To address these challenges, we propose a novel generative-based rehearsal strategy, called Dirichlet Continual Learning (DCL), specifically tailored for CL in NLP. In contrast to conventional approaches that employ the Gaussian latent variable in Conditional Variational Autoencoders (CVAE), DCL harnesses the flexibility and versatility of the Dirichlet distribution to model the latent prior variable. This unique feature empowers DCL to effectively capture and represent sentence-level features from previous tasks, laying a solid foundation for generating high-quality pseudo samples. Moreover, we introduce Jensen-Shannon Knowledge Distillation (JSKD), a robust logit-based knowledge distillation method that enhances knowledge transfer during the generation of pseudo samples. By accurately measuring the similarity of the teacher model learned from previous tasks and the student model learned from the current task, JSKD facilitates the transfer of accumulated knowledge from previous tasks to the current task. This approach augments the effectiveness of the rehearsal process, ensuring that the generated samples retain and reflect the task-specific knowledge acquired by the model.

We summarize our main contributions as follows:

- We propose a novel generative-based rehearsal method, which effectively addresses the issue of forgetting and diversity by employing the Dirichlet latent variable within the framework of Conditional Variational Autoencoders (CVAE). This approach enables us to approximate the real data distribution more accurately, thereby improving the quality of generated pseudo samples.
- We introduce Jensen-Shannon Knowledge Distillation (JSKD), a new logit-based knowledge distillation strategy that enhances the transfer of knowledge between teacher and student models. JSKD facilitates more effective and robust knowledge transfer, leading to improved performance in the generation of pseudo samples.
- Through extensive experiments on two typical tasks in task-oriented dialogue systems, we demonstrate the remarkable performance improvement achieved by our proposed DCL compared to state-of-the-art baselines for CL in NLP. DCL achieves performance close to the upper bound of multi-task learning, showcasing its effectiveness. Most notably, we outperform the state-of-the-art generative-based rehearsal method, Prompt Conditioned VAE for Lifelong Learning (PCLL), in all metrics. Specifically, in Intent Detection, our DCL

method achieves a significant 3.48% accuracy improvement and a 4.22% Learning Curve Area (LCA) improvement. Similarly, in Slot Filling, DCL demonstrates a notable 2.89% accuracy improvement and a 6.08% LCA improvement.

## 2 RELATED WORK

We highlight related work in the following two subsections.

### 2.1 CONTINUAL LEARNING

CL can be categorized into three main strategies:

- *Regularization* methods aim to strengthen previous knowledge by imposing constraints on important parameters and incorporating regularization terms into the loss function. One notable approach is Elastic Weight Consolidation (EWC) [Kirkpatrick et al., 2017], which identifies crucial parameters and prevents their updates, thereby preserving performance on previous tasks. Additionally, the Synaptic Intelligence (SI) model [Zenke et al., 2017] dynamically computes per-synapse consolidation strength throughout the learning trajectory. Memory Aware Synapses (MAS) [Aljundi et al., 2018] determine parameter importance in an online and unsupervised manner. Learning without Memorizing (LwM) [Dhar et al., 2019] utilizes attention distillation loss to support the progressive learning of new classes, making it effectively preserves information on base classes when incorporating new classes. However, the accumulation of multiple regularizers may overly constrain network parameters, potentially hindering the learning of new tasks.
- *Architectural* approaches usually modify the network structure to capture task-specific features and mitigate catastrophic forgetting. PathNet [Fernando et al., 2017] introduces dynamic pathway evolution, allowing the model to learn task-specific paths through a shared network. CL with GANs [Seff et al., 2017] employs adversarial training to achieve a balance between learning new tasks and retaining knowledge from previous tasks. Piggyback GAN [Zhai et al., 2020] shares filter between tasks, enabling high-quality generation with fewer parameters while preserving performance on previous tasks. However, their task-focused approach may overlook effective knowledge transfer between old and new tasks.
- *Rehearsal* methods, utilized to sustain performance by leveraging samples from previous tasks, can be classified into two categories: store-based rehearsal and generative-based rehearsal. Store-based rehearsal methods, such as Replay in Deep Learning [Hayes et al., 2021] and Gradient-based Memory Editing (GMED) [Jin et al., 2021], rely on episodic memory

to store examples from previous tasks. For instance, iCaRL [Rebuffi et al., 2017] incorporates a herding-based step to select representative samples and alleviate the catastrophic forgetting problem. Gradient Episodic Memory (GEM) [Lopez-Paz and Ranzato, 2017] stores and replays important exemplars from previous tasks while learning new ones. Experience replay (ER) for CL [Rolnick et al., 2019] continuously trains the model using batch gradient descent by sampling examples from both the current task and the episodic memory. On the other hand, generative-based rehearsal methods, such as CL with deep generative replay [Shin et al., 2017], rely on generative adversarial networks (GANs) [Goodfellow et al., 2014] to create synthetic samples. ReMix [Mi et al., 2020b] generates pseudo samples by applying Mixup [Zhang et al., 2017] to samples from previous tasks. However, store-based rehearsal methods can be inefficient and require increased memory demands, while generative-based rehearsal methods may suffer from limited diversity, fluency, or poor alignment with the designated task.

## 2.2 CONTINUAL LEARNING IN NLP

Within the context of *CL in NLP*, researchers have explored various strategies to address the challenges associated with evolving language tasks:

- *Regularization* Adaptively Regularized Prioritized Exemplar Replay (ARPER) [Mi et al., 2020a] presents the initial attempt to explore a practical continual learning configuration for Natural Language Generation (NLG) by incorporating prioritized exemplar replay and adaptive regularization based on Elastic Weight Consolidation (EWC). Specifically, ARPER prioritizes representative and diverse utterances in exemplar selection, aiming to comprehensively cover information from previous tasks.
- *Architectural* Adapter-based Continual Learning (AdapterCL) [Madotto et al., 2021] is an architectural approach that places residual adapters [Houlsby et al., 2019] atop the transformer layer to approximate each task in a continual learning setting. Continual Prompt Tuning (CPT) [Zhu et al., 2022] ensures non-forgetting and bidirectional knowledge transfer in a parameter-efficient dialog system. It utilizes techniques such as prompt learning, memory replay, and query fusion to facilitate continual learning. Semi-Supervised Lifelong Learning (SSLL) [Zhao et al., 2022b] integrates both labeled and unlabeled data for sequentially arriving tasks in a continual learning setting. It incorporates specialized modules to mitigate forgetting and harness the potential of unlabeled data. Adaptive Continual Modeling (ACM) [Zhang et al., 2022] adopts a two-stage method to achieve efficient continual sequence generation. It

dynamically adds or reuses modules based on task similarity and employs pseudo rehearsal for effective knowledge transfer, outperforming existing baselines in continual learning scenarios. GRACE [Hartvigsen et al., 2022], a novel model editing method, enables thousands of sequential edits on deployed language models to address performance decay over time without significant degradation.

- *Rehearsal* Language Model with Adaptive Learning (LAMOL) [Sun et al., 2019] is a rehearsal method based on continual sequence generation. The generative-based rehearsal approach employed by LAMOL does not require memory to store previous samples. L2KD [Chuang et al., 2020] improves LAMOL by assigning an extra teacher for each new task to perform knowledge distillation PCLL [Zhu et al., 2022] adopts a CVAE to generate pseudo samples from past tasks, which are then used for continual learning.

The remarkable performance of PCLL motivates us to explore further in this paper.

## 3 METHOD

### 3.1 PROBLEM FORMULATION

A CL model in NLP aims to learn a stream of NLP tasks sequentially, $T = \{T_1, \cdots, T_N\}$, where $N$ is the number of tasks, which can be potentially infinite. For task $n$, denoted by $T_n$, its data $\mathcal{D}_n = \{(x_i, y_i)\}_{i=1}^{N_n}$ are drawn from an underlying data distribution. Here, $x_i$ denotes an input utterance, and $y_i$ denotes the output label. In Intent Detection, a typical example can be ("I need you to get me a flight booked from Houston to Miami on United Airlines", "book flight"). Meanwhile, in Slot Filling, a typical example is {*"utterance: "how many comedy movies starring kevin costner have come out in the year 2000", "GENRE: comedy; ACTOR: kevin costner"*}. Our goal is to train a model capable of excelling in all encountered tasks while minimizing the extent of forgetting.

### 3.2 OVERVIEW

Figure 1 depicts the key components of our proposed method, called Dirichlet Continual Learning (DCL). DCL comprises the pseudo rehearsal module and the Language Model (LM) training module. Prior to training the current task, we utilize CVAE with the Dirichlet distribution to model the latent variable, generating pseudo samples from previous tasks. DCL then continues training using both the pseudo data and real data from the current task, enhancing model robustness through the employment of JSKD. In the CVAE framework, the encoder and decoder utilize a pre-trained language model, such as GPT-2, with separate

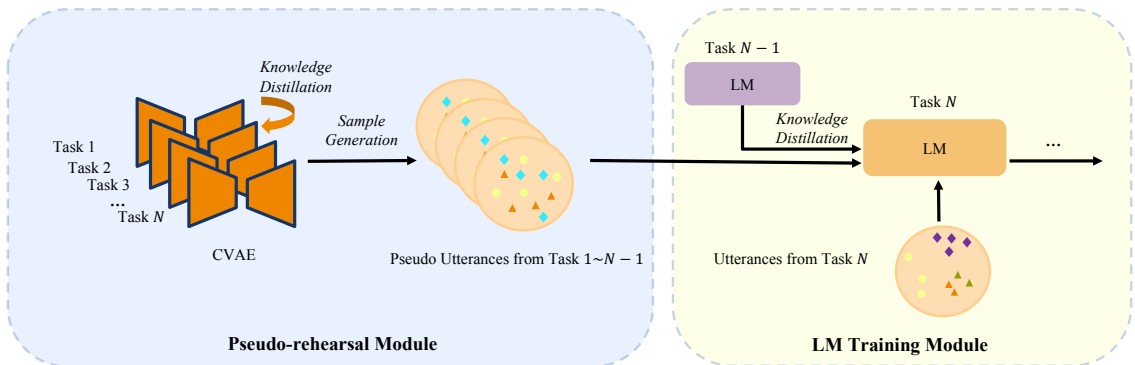

Figure 1: DCL consists of two key modules: the pseudo rehearsal module and the LM training module. The DCL model follows two main steps: (1) During the training of Task $N$, the pseudo rehearsal module uses CVAE with Dirichlet latent variable to generate pseudo samples from Task 1 to Task $N-1$. (2) These pseudo samples are combined with the training data from Task $N$ to further train DCL, with the inclusion of JSKD to ensure robust training.

parameters. Importantly, the LM shares parameters with the decoder of CVAE, enabling the construction of a unified model to address the CL problem.

### 3.3 MODULES

**LM Training Module** For data sample $(x_i, y_i) \in \mathcal{D}_n$ in task $T_n$, given an input utterance $x_i$, LM aims to generate the corresponding output $y_i$. To achieve task-dependent generation, we define specific prefix prompt $P_n$ and postfix prompt $P_n^*$ for task $T_n$. These prompts are concatenated with the input utterance, resulting in the augmented input $g(x_i) = P_n \oplus x_i \oplus P_n^*$, where $\oplus$ denotes word concatenation. The details of $P_n$ and $P_n^*$ can be found in Appendix A. The LM for task $T_n$ is optimized based on $g(x_i)$ which carries the task-specific information. The LM is optimized by minimizing the loss which is defined by $\mathcal{L}_{\mathrm{LM}}$ as follows:

$$-\sum_{(x_i, y_i) \in \mathcal{D}_n} \log p_\theta(g(x_i), y_i) + \log p_\theta(y_i|g(x_i)). \quad (1)$$

**Gaussian-guided Pseudo Rehearsal Module** Following the setting of PCLL, we employ CVAE to model the distribution of high-dimensional data $x$ using lower-dimensional latent variable $z$ [Agarwal et al., 2023]. Let $z$ be a continuous variable representing the sentence-level features of input utterance $x$, and we take the task ID as $c$. The generative process involves an encoder $q_\phi(z|x, c)$ mapping $x$ to approximate the true posterior $p(z|x, c)$. The latent variable is sampled from $q_\phi(z|x, c)$, and a decoder $p_\theta(x|z, c)$ reconstructs $x$. The CVAE is trained to maximize the log-likelihood $\log p(x|c)$, which is tractable by maximizing the

evidence lower bound (ELBO) $L(\theta, \phi; x, c)$:

$$\underbrace{L(\theta, \phi; x, c)}_{-\mathcal{L}_{\mathrm{CVAE}}^G} = -\lambda \underbrace{\mathrm{KL}(q_\phi(z|x, c)||p_\theta(z|c))}_{\mathcal{L}_{\mathrm{KL}}^G}$$
$$- \underbrace{(-\mathbb{E}_{q_\phi(z|x, c)}[\log p_\theta(x|z, c)])}_{\mathcal{L}_{\mathrm{Rec}}}$$
$$\leq \log p(x|c), \quad (2)$$

where $\mathcal{L}_{\mathrm{CVAE}}^G$ denotes the CVAE loss which is the negative of ELBO. $\mathcal{L}_{\mathrm{KL}}^G$ is the Kullback–Leibler (KL) loss measuring the distribution distance of Gaussian latent. $\mathcal{L}_{\mathrm{Rec}}$ represents the reconstruction loss. $\theta$ is the model parameter, $p_\theta(z|c)$ is the prior distribution of $z$, and $\lambda$ is the dynamic KL weight, gradually increasing from 0 to 1 via the annealing technique, to mitigate the KL-vanishing as proposed by Bowman et al. [2016]. The CVAE is trained by minimizing $\mathcal{L}_{\mathrm{CVAE}}^G$.

**Dirichlet-guided Pseudo Rehearsal Module** In contrast to conventional CVAE, where the latent variable $z \sim \mathcal{N}(\boldsymbol{\mu}, \boldsymbol{\Sigma})$ is sampled from a symmetric multivariate Gaussian in a continuous space, our DCL samples the latent variable $z \sim \mathrm{Dir}(\boldsymbol{\alpha})$ from a Dirichlet distribution in a discrete space, with $\boldsymbol{\alpha}$ as the parameter vector. By leveraging the versatile mathematical properties of the Dirichlet distribution, our DCL model better approximates the actual token distribution, which can exhibit concave, convex, symmetrical, or asymmetrical behavior. Since the Cumulative Distribution Function (CDF) of the Dirichlet distribution is unknown, we cannot employ the inverse CDF for sampling the Dirichlet latent variable. Therefore, we use rejection sampling, following the suggestion by [Jankowiak and Obermeyer, 2018]. DCL then replaces $\mathcal{L}_{\mathrm{CVAE}}^G$ by $\mathcal{L}_{\mathrm{CVAE}}^D$ and aims to train the model by minimizing the following loss:

$$\mathcal{L}_{\mathrm{DCL}} = \mathcal{L}_{\mathrm{LM}} + \underbrace{\lambda \mathcal{L}_{\mathrm{KL}}^D + \mathcal{L}_{\mathrm{Rec}}}_{\mathcal{L}_{\mathrm{CVAE}}^D}, \quad (3)$$

where $\mathcal{L}_{\mathrm{KL}}^D$ is the KL loss on the Dirichlet latent and is expressed as [Zeng et al., 2019]:

$$\log\Gamma(\sum_{k=1}^{K}\alpha_k) - \sum_{k=1}^{K}\log\Gamma(\alpha_k) - \log\Gamma(\sum_{k=1}^{K}\beta_k) \quad (4)$$

$$+ \sum_{k=1}^{K}\log\Gamma(\beta_k) + \sum_{k=1}^{K}(\alpha_k - \beta_k)(\psi(\alpha_k) - \psi(\sum_{k=1}^{K}\alpha_k)),$$

where $\alpha$ and $\beta$ represent the parameters of the Dirichlet distributions $q_\phi(z|x,c)$ and $p_\theta(z|c)$, respectively. $K$ denotes the dimension of $z$. $\Gamma$ is the gamma function and $\psi$ is the Digamma function.

### 3.4 KNOWLEDGE DISTILLATION

To address the issue of potential drift in the pseudo data distribution and the negative impact of noise when combining generated pseudo samples with real training data, we leverage knowledge distillation (KD) techniques [Chuang et al., 2020, Mi et al., 2020a, Zhao et al., 2022a, Chen et al., 2023] to safeguard DCL against the adverse effects of noisy pseudo data.

In DCL, the student model $f_{\theta_n}$ is trained on task $T_n$, while the teacher model $f_{\theta_{n-1}}$ is trained on task $T_{n-1}$. We distill the knowledge from $f_{\theta_{n-1}}$ to $f_{\theta_n}$ using the pseudo data, minimizing the knowledge distillation loss. As training progresses, the roles are switched, with the $T_n$ model becoming the teacher for $T_{n+1}$. This iterative role-switching enables the accumulation of cumulative knowledge from previous tasks, ensuring effective CL.

**KD via KL Divergence** Given a training sample $(x_i, y_i)$, the goal is to minimize the cross-entropy between the probability distributions produced by the teacher and student models [Chuang et al., 2020]. The knowledge distillation loss is:

$$\mathcal{L}_{\mathrm{KD}} = \alpha \cdot \mathcal{L}_{\mathrm{KL}}(S_i, T_i) \cdot \tau^2 + (1-\alpha) \cdot \mathcal{L}_{\mathrm{CE}}(S_i, y_i), \quad (5)$$

where $S_i = f_{\theta_n}(x_i)$ and $T_i = f_{\theta_{n-1}}(x_i)$ are output logits of the student and teacher model for the input $x_i$, respectively. $\tau$ is the temperature to soften the teacher's prediction while $\mathcal{L}_{\mathrm{CE}}(S_i, y_i)$ quantifies the cross-entropy loss between student prediction and the ground truth label $y_i$. $\mathcal{L}_{\mathrm{KL}}$ implicitly prevents the parameters of the student model from straying too far away from the ones of the teacher model. The first term denotes a soft target while the second term is a hard target. $\alpha \in [0,1]$ trades off two terms.

**KD via JS Divergence** Previous methods, such as [Hinton et al., 2015], employ KL divergence to measure the distribution similarity between student and teacher models. However, KL divergence is sensitive to outliers, making it less robust in scenarios with noisy or sparse data. To address this limitation, we introduce Jensen-Shannon (JS)

divergence as a more stable alternative, incorporating a symmetric term. Leveraging this, we propose Jensen-Shannon Knowledge Distillation (JSKD), a novel logit-based method, which achieves remarkable performance in knowledge distillation.

**JS Divergence** Given distributions $p$ and $q$, JS divergence [Lin, 1991] is defined by:

$$\mathcal{L}_{\mathrm{JS}}(p \,\|\, q) = \tfrac{1}{2}[\mathcal{L}_{\mathrm{KL}}\left(p \,\|\, \tfrac{1}{2}(p+q)\right) + \mathcal{L}_{\mathrm{KL}}\left(q \,\|\, \tfrac{1}{2}(p+q)\right)]. \quad (6)$$

JS divergence offers advantages over KL divergence: a) its symmetry ensures consistent values regardless of comparison order, making it ideal for measuring distribution similarities; b) JS divergence is bounded in the range $[0, 1]$, while KL divergence spans $[0, +\infty)$; c) Unlike the asymmetric KL divergence, JS divergence symmetrically measures the similarity between two probability distributions, with values ranging from 0 (for identical distributions) to 1 (for distributions with no shared support).

These properties make JS divergence more suitable for knowledge distillation compared to KL divergence, particularly KL divergence may yield infinite value when one sample appears exclusively in one task distribution.

Based on the above discussions, we propose JSKD to measure the distance between the student and teacher models, improving model robustness. The JSKD loss is defined as:

$$\mathcal{L}_{\mathrm{KD}} = \alpha \cdot \mathcal{L}_{\mathrm{JS}}(S_i, T_i) \cdot \tau^2 + (1-\alpha) \cdot \mathcal{L}_{\mathrm{CE}}(S_i, y_i). \quad (7)$$

Specifically, we adopt the model trained on $T_{n-1}$ as the teacher model and the current task $T_n$ as the student model. Incorporating knowledge distillation, the $\mathcal{L}_{\mathrm{Rec}}$ and $\mathcal{L}_{\mathrm{LM}}$ for task $T_n$ is modified as:

$$\mathcal{L}_{\mathrm{Rec}}^{\mathrm{JS}} = \alpha \cdot \mathcal{L}_{\mathrm{JS}}(l_c, l_c^*) \cdot \tau^2 + (1-\alpha) \cdot \mathcal{L}_{\mathrm{CE}}(l_c, y_i), \quad (8)$$

$$\mathcal{L}_{\mathrm{LM}}^{\mathrm{JS}} = \alpha \cdot \mathcal{L}_{\mathrm{JS}}(l_l, l_l^*) \cdot \tau^2 + (1-\alpha) \cdot \mathcal{L}_{\mathrm{CE}}(l_l, y_i), \quad (9)$$

where $l_c$ and $l_l$ are the logits output of CVAE and LM of task $T_n$, respectively. $l_c^*$ and $l_l^*$ represent the logits output of task $T_{n-1}$, and $y_i$ signifies the ground truth.

Hence, our DCL with JSKD is trained by minimizing the following loss:

$$\mathcal{L}_{\mathrm{DCL}}^{\mathrm{JS}} = \mathcal{L}_{\mathrm{LM}}^{\mathrm{JS}} + \lambda\mathcal{L}_{\mathrm{KL}}^D + \mathcal{L}_{\mathrm{Rec}}^{\mathrm{JS}}. \quad (10)$$

We emphasize that $\mathcal{L}_{\mathrm{KL}}^D$ is the KL divergence in Eq. (4) to evaluate the distance between the assumed Dirichlet data distribution and the real distribution. It is different from the $\mathcal{L}_{\mathrm{KL}}$ where it evaluates the distance between the student and teacher models in cross-task knowledge distillation.

## 4 EXPERIMENTS

### 4.1 DATASETS

Based on the setup and pre-processed datasets described in PCLL [Zhao et al., 2022a], as detailed in Appendix B, we conduct experiments to simulate continual learning (CL) for two NLP tasks in task-oriented dialogue systems: Intent Detection and Slot Filling. In Intent Detection, we utilize six datasets annotated with intents: HWU [Liu et al., 2021], BANKING [Casanueva et al., 2020], CLINC [Larson et al., 2019], SNIPS [Coucke et al., 2018], AITS [Hemphill et al., 1990], and TOP [Gupta et al., 2018] datasets. Notably, the TOP dataset is divided into three distinct subsets: TOP-S1, TOP-S2, and TOP-S3. These subsets, along with the other five datasets, constitute a total of eight tasks used to evaluate CL in the intent detection experiment. In Slot Filling, we adopt the SNIPS, AITS, DSTC [Rastogi et al., 2020], MIT-MOVIE, and MIT-RESTAURANT datasets. These datasets yield a total of five tasks to evaluate CL in the slot filling experiment. For a fair comparison, these tasks are learned in six different orders, as depicted in Appendix C, and the average performances of these orders are reported.

### 4.2 COMPARED METHODS

We compare our DCL with eleven competitive baselines: (1) **Fine-tune** directly fine-tunes GPT-2 on the task stream without preventing catastrophic forgetting (CF); (2-3) Two typical *regularization* methods are **EWC** [Kirkpatrick et al., 2017] and **MAS** [Aljundi et al., 2018], which penalize changes of important parameters to mitigate forgetting; (4-6) Three *architectural* methods include **HAT** [Serra et al., 2018] with task-based hard attention, **CTR** [Ke et al., 2021] which inserts continual learning plug-ins into BERT, and **AdapterCL** [Madotto et al., 2021] with task-specific residual adapters; (7-11) Four *rehearsal* methods consist of two variants of LAMOL [Sun et al., 2019], **LAMOL-g** and **LAMOL-t** for global incorporation and task-specific tokens respectively, **ER** [Rolnick et al., 2019] which preserves previously seen real samples for replay, **L2KD** [Chuang et al., 2020] which performs knowledge distillation with an extra teacher, and **PCLL** [Zhao et al., 2022a], the current state-of-the-art method that applies prompt conditioned VAE for lifelong learning; (12) Additionally, we evaluate the model performance when all tasks are trained simultaneously in a multi-task learning setting (**Multi**), which serves as the upper bound in the comparison.

### 4.3 EXPERIMENTAL SETTINGS

The experiments are conducted on an NVIDIA A100-80GB GPU, utilizing the Adam optimizer for both tasks. For Intent Detection, the following parameter settings are applied

Table 1: Comparison results of DCL and baselines. Scores of baselines are reported in [Zhao et al., 2022a] and the performance of Multi is the upper bound. The best results are highlighted in bold.

| Models | Intent Detection (%) | | Slot Filling (%) | |
|---|---|---|---|---|
| | Score ↑ | LCA ↑ | Score ↑ | LCA ↑ |
| Finetune | 14.09 | 28.76 | 15.38 | 19.55 |
| EWC | 14.16 | 28.34 | 15.67 | 19.51 |
| MAS | 14.15 | 28.61 | 15.59 | 19.37 |
| HAT | 73.92 | 73.03 | 61.99 | 67.33 |
| CTR | 67.44 | 71.11 | 63.84 | 67.28 |
| AdapterCL | 81.15 | 75.60 | 75.60 | 48.47 |
| L2KD | 35.22 | 61.78 | 44.16 | 39.94 |
| LAMOL-g | 50.30 | 60.67 | 45.12 | 38.03 |
| LAMOL-t | 51.81 | 67.97 | 44.83 | 37.58 |
| ER | 78.19 | 78.19 | 44.95 | 39.32 |
| PCLL | 90.25 | 88.82 | 74.48 | 68.41 |
| **DCL** | **93.73** | **93.04** | **77.37** | **74.49** |
| Multi (Upper Bound) | 96.25 | N/A | 80.80 | N/A |

due to their good performance: batch size: 32, learning rate: 5e-5, pseudo-sample rate: 0.2, $z$'s dimension: 128, maximum context length: 256, and the number of epochs: 5. In knowledge distillation (KD), $\alpha$ was set to 0.9 and $\tau$ to 2.0. Unless specified otherwise, the parameters used in Slot Filling are the same as those in Intent Detection, except for $z$'s dimension (set to 512), maximum context length (set to 50), the number of epochs (set to 10), and the KD parameters $\alpha$ (set to 1.0) and $\tau$ (set to 2.0). The training time for DCL on Intent Detection using a single GPU is approximately 5 hours, while for Slot Filling, it is around 6 hours.

### 4.4 EVALUATION METRICS

We adopt the accuracy score and macro-averaged F1 score to evaluate the performance of Intent Detection and Slot Filling, respectively. Following [Lopez-Paz and Ranzato, 2017, Chaudhry et al., 2018], we employ the following two metrics to evaluate the performance: (1) **Average Score (Score)** [Lopez-Paz and Ranzato, 2017] defines the average accuracy on all tasks after the final task has been learned: $\text{Score} = \frac{1}{T}\sum_{i=1}^{T} R_{T,i}$, where $R_{i,j}$ denotes the evaluation metric on task $t_j$ after training on task $t_i$. (2) **Learning Curve Area (LCA)** [Chaudhry et al., 2018] is the area under a learning curve to indicate a model's performance in a sequence of tasks: $\text{LCA} = \int_0^T P(t)dt$, where $P(t)$ is the average model performance at step $t$ across all already-learnt tasks, and $T$ is the total number of steps. Higher LCA values indicate better performance.

### 4.5 MAIN RESULTS

Table 1 presents the performance of our proposed DCL model compared to the baselines. DCL demonstrates a sig-

nificant improvement over all baselines in both tasks:

- DCL surpasses the state-of-the-art model, PCLL, by a wide margin, achieving superior results across all evaluation metrics. In Intent Detection, DCL achieves a remarkable 3.48% increase in accuracy score and a 4.22% improvement in LCA. Similarly, in Slot Filling, DCL achieves a notable 2.89% increase in F1 score and a substantial 6.08% improvement in LCA. These impressive enhancements are attributed to the effective utilization of Dirichlet-guided pseudo rehearsal and JSKD techniques. By leveraging these components, DCL generates a more diverse and representative set of examples, leading to further optimization of model performance. The generation of diverse and representative examples is crucial for effectively capturing task-specific information, particularly when the number of available pseudo samples is limited.

- DCL achieves performance that is comparable to the upper bound in Multi-task learning (**Multi**), with only a slight lag of 2.52% in accuracy for Intent Detection and 3.43% in F1 score for Slot Filling. This slight difference in performance can be attributed to variations in the amount of data and the realism of the samples used for evaluation.

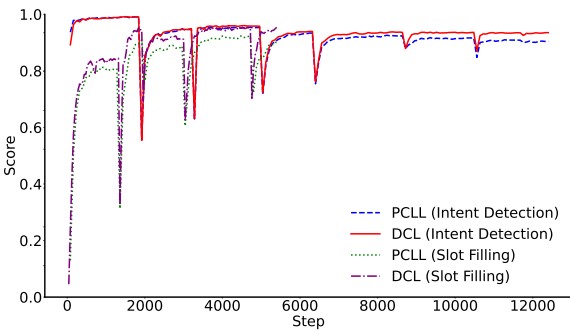

Figure 2: Learning curves of DCL and PCLL.

### 4.6 ABLATION STUDY

**Learning curves of DCL vs. PCLL** Figure 2 presents the learning curve of average scores to gain deeper insights into DCL and PCLL. The significant drop in accuracy observed during task switching demonstrates the challenge of catastrophic forgetting. However, DCL proves to be more effective in mitigating this issue, as evidenced by its higher accuracy compared to PCLL. Specifically, DCL outperforms PCLL in Intent Detection after 6,000 steps, whereas it achieves a similar superiority over PCLL in Slot Filling after around 1,000 steps. This trend suggests that as the model encounters a more diverse range of tasks, the benefits of DCL become more pronounced, highlighting the promising nature of our approach.

Table 2: Results of DCL with KL knowledge distillation and PCLL in two tasks. DCL outperforms PCLL across all metrics.

| Models | Intent Detection (%) | | Slot Filling (%) | |
|---|---|---|---|---|
| | Score ↑ | LCA ↑ | Score ↑ | LCA ↑ |
| PCLL | 90.25 | 88.82 | 74.48 | 68.41 |
| DCL (with KL) | **92.83** | **91.32** | **76.42** | **73.76** |

Table 3: F1 score(%) and LCA (%) of Slot Filling on six orders with KL and JS knowledge distillation.

| Orders | DCL (with KL) | | DCL (with JS) | |
|---|---|---|---|---|
| | Score ↑ | LCA ↑ | Score ↑ | LCA ↑ |
| order 0 | 80.26 | 73.82 | 81.39 | 74.66 |
| order 1 | 80.31 | 74.82 | 80.94 | 75.23 |
| order 2 | 74.04 | 70.01 | 74.77 | 70.73 |
| order 3 | 77.00 | 76.28 | 78.03 | 76.69 |
| order 4 | 72.98 | 70.10 | 74.53 | 70.99 |
| order 5 | 73.92 | 77.50 | 74.56 | 78.61 |
| Mean | 76.42 | 73.76 | **77.37** | **74.49** |

**Gaussian vs. Dirichlet-guided Rehearsal Module.** We conduct a comparative analysis of DCL and PCLL in Intent Detection and Slot Filling to evaluate the impact of a Dirichlet-guided rehearsal module. To ensure a fair comparison, DCL incorporates KL knowledge distillation, similar to PCLL. The results presented in Table 2 demonstrate that DCL, with the Dirichlet-guided rehearsal module, consistently outperforms PCLL, which utilizes a Gaussian-guided module, across all evaluation metrics in both tasks. These findings indicate that the Dirichlet distribution provides a more effective approximation of the true data distribution, leading to better performance.

**KL Knowledge Distillation vs. JSKD.** We evaluate the impact of JSKD on the performance of DCL and compare it with DCL equipped with KL Knowledge Distillation. Due to space limitations, we only report the performance in Slot Filling under different learning orders, as summarized in Table 3, where the performance in Intent Detection follows similar trends and we report it in Table 4. Our findings consistently demonstrate that DCL equipped with JS knowledge distillation significantly outperforms DCL with KL knowledge distillation across all task learning orders with 99% confidence level on the paired $t$-test. Notably, we observe significant improvements ranging from 0.63% to 1.55% in the F1 score and 0.41% to 1.11% in LCA. This conclusion also holds for Intent Detection.

**Number of Pseudo Samples.** To assess the impact of the number of pseudo samples on the proposed approach, we experiment with different ratios of pseudo samples in DCL. Table 5 compares the performance of PCLL to DCL with ratios of 0.1, 0.2, 0.4, and 0.5. Results about different ratios of PCLL can be referred to Zhao et al. [2022a]. Remarkably, even with fewer pseudo samples incorporated during training, DCL with a ratio of 0.1 surpasses PCLL with a

Table 4: ACC score(%) and LCA (%) of Intent Detection on six orders with KL and JS knowledge distillation.

| Orders | DCL (with KL) | | DCL (with JS) | |
|---|---|---|---|---|
| | Score ↑ | LCA ↑ | Score ↑ | LCA ↑ |
| order 0 | 92.76 | 91.27 | 93.52 | 92.69 |
| order 1 | 92.76 | 88.49 | 93.64 | 90.45 |
| order 2 | 92.56 | 93.36 | 93.74 | 95.47 |
| order 3 | 92.91 | 91.53 | 93.83 | 93.34 |
| order 4 | 92.68 | 89.92 | 93.79 | 92.04 |
| order 5 | 93.33 | 93.32 | 93.85 | 94.22 |
| Mean | 92.83 | 91.32 | **93.73** | **93.04** |

Table 5: Comparison result of PCLL with pseudo samples ratio of 0.2 and DCL with a different number of pseudo samples numbers ranging from 0.1 to 0.5 in Intent Detection.

| Model | Ratio | Score | LCA |
|---|---|---|---|
| PCLL | 0.05 | 84.09 | 89.54 |
| PCLL | 0.2 | 90.25 | 88.82 |
| PCLL | 0.5 | 91.02 | 91.44 |
| PCLL | 1.0 | 91.31 | 91.77 |
| DCL | 0.1 | 91.66 | 91.83 |
| DCL | 0.2 | 93.73 | **93.04** |
| DCL | 0.4 | 93.97 | 92.76 |
| DCL | 0.5 | **94.23** | 92.82 |

ratio of 0.2 in both metrics. This demonstrates the superiority of DCL. Furthermore, we observe that performance improves with an increasing number of pseudo samples. This can be attributed to the fact that more samples carry more information, thereby enhancing the capabilities of the model.

**Evaluating Pseudo Samples Quality.** To compare the quality of the generated pseudo samples in different baselines with our proposed model, we utilize **Dist-n** [Li et al., 2015] to assess pseudo samples. Dist-n measures the proportion of distinct n-grams in the generated pseudo samples. A higher Dist-n value corresponding to larger pseudo sample diversity is preferred where the samples are more distinct. We employ Dist-1, Dist-2, Dist-3, and Dist-4 to analyze the quality of generated samples completely.

Given the limited number of pseudo samples, the quality of our exemplars is crucial to preserve the performance of previous tasks. We aim to carefully select representative and diverse utterances instead of generic and similar ones. Table 6 summarizes the Dist-n results. Notably, DCL achieves higher distinct scores compared to other methods, indicating that DCL-generated pseudo samples exhibit larger diversity. This suggests that pseudo samples created by DCL are more similar to real samples.

**Dimension of Latent Variable.** Table 7 highlights the impact of the dimension for $z$ in DCL. Notably, DCL employing JSKD with a small dimension of 8 outperforms DCL using KL with a larger dimension of 128. This suggests that

Table 6: Distinct scores for generated pseudo samples. A higher Dist-n score means higher diversity.

| Model | Dist-1 | Dist-2 | Dist-3 | Dist-4 |
|---|---|---|---|---|
| LAMOL-g | 0.0602 | 0.2466 | 0.4489 | 0.6178 |
| LAMOL-t | 0.1758 | 0.4733 | 0.6837 | 0.8090 |
| PCLL | 0.2836 | 0.6566 | 0.8369 | 0.9221 |
| **DCL** | **0.3092** | **0.7019** | **0.8708** | **0.9389** |
| Real Sample | 0.4000 | 0.7972 | 0.9255 | 0.9717 |

Table 7: Intent detection of DCL (with JS) and DCL (with KL) for different latent variable dimensions.

| Models | Score | LCA |
|---|---|---|
| DCL (with KL, z = 128) | 92.83 | 91.32 |
| DCL (with JS, z = 8) | 93.51 | **93.11** |
| DCL (with JS, z = 128) | **93.73** | 93.04 |

the DCL model can generate high-quality pseudo samples even with smaller dimensions and less encoded information, resulting in improved accuracy. These results demonstrate the superiority of the Dirichlet latent over the Gaussian counterpart. However, it is important to note that DCL using JSKD with a latent dimension of 8 exhibits degraded performance compared to that of 128. This is attributed to the reduced information capacity in the scenario with a smaller dimension in $z$.

**Llama2 as a Backbone.** To evaluate the scalability of DCL in large language models (LLMs), we employ Llama 2-7B as the backbone to conduct experiments on Intent Detection. The results at order 0 presented in Table 8 show that DCL outperforms PCLL with a significant improvement of 5.72%, demonstrating its efficiency and scalability. Fine-tuning of both DCL and PCLL models using Lora on a single GPU took approximately 80 hours. However, the performance is noticeably worse than fine-tuning with GPT-2 due to the limited amount of training data available.

## 4.7 CASE STUDY

Table 9 illustrates the pseudo samples generated by DCL, PCLL, and real samples (Golden) from Intent Detection. PCLL encounters challenges in accurately generating utterances, as seen in examples like *"Do they have a lot of miles on this road"* for the intent of *"mpg"* (miles per gallon). Similar issues arise with the utterance *"Do you know how much my new credit card is worth?"*. In contrast, DCL consistently produces appropriate utterances with the corresponding intents. It showcases the superior capability of DCL in generating enhanced pseudo samples that align more consistently with the task distribution.

Table 8: Performance of DCL and PCLL using Llama2 as backbone on Intent Detection.

| Models | Score |
|--------|-------|
| PCLL (Llama2) | 66.69 |
| DCL (Llama2) | **72.41** |

Table 9: Illustration of generated pseudo samples by PCLL and DCL with the Ground Truth (Golden).

| Models | Input Utterance | Output |
|--------|-----------------|--------|
| Golden | what's the fuel economy of my car. | mpg |
| PCLL | Do they have a lot of miles on this road? | mpg |
| DCL | What is the mpg of this car? | mpg |
| Golden | What is the expiration date on my card? | expiration date |
| PCLL | Do you know how much my new credit card is worth? | expiration date |
| DCL | Can you check my expiration month? | expiration date |

## 5 CONCLUSION

In this paper, we propose DCL, a generative-based rehearsal method, to address CF in CL. Our DCL leverages a Dirichlet distribution-based CVAE, which offers greater flexibility in modeling utterance-level characteristics and generates more realistic pseudo samples compared to the conventional Gaussian-based CVAE. Furthermore, we propose a robust JS divergence-based knowledge distillation method to enhance knowledge transfer between tasks. Extensive experiments validate the effectiveness of our DCL, demonstrating its superiority over state-of-the-art methods.

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

# Supplementary Material

**Min Zeng**[1]  **Haiqin Yang**[2]  **Wei Xue**[1]  **Qifeng Liu**[1]  **Yike Guo**[*1]

[1]Hong Kong University of Science and Technology
[2]International Digital Economy Academy (IDEA)

## A   PROMPT EXAMPLES

The prompt for **intent detection** is *"For an utterance from the ID task, $x$ has the following intent"*.

- For example, when training on a *BANKING* task, the input $x$ is *"Please tell me how to link the card?"*, and the modified $x$ is *"For an utterance from the BANKING task, "Please tell me how to link the card?" has the following intent"*. Thus the output $y$ is its corresponding intent *"card linking"*.

The prompt for **slot filling** is *"In the ID task, if there are any slots and values, what are they in this sentence: $x$? Answer: "*.

- For example, when training on a *DSTC* task, the input is *"I'm planning on leaving the 11th of this month and there are 3 people traveling."*, and the modified $x$ is *"In the DSTC task, if there are any slots and values, what are they in this sentence: "I'm planning on leaving the 11th of this month and there are 3 people traveling."? Answer: "*. Thus the output $y$ is its corresponding slot-value pairs returns *"leaving date: 11th of this month"*.

## B   DATASET DETAILS

As reported in PCLL [Zhao et al., 2022a], we highlight the details of the datasets as follows:

- **ATIS** dataset consists of audio recordings and manual transcriptions capturing user inquiries about flight information in automated airline travel inquiry systems, featuring 17 distinct intent categories and serving as a benchmark for evaluating natural language processing systems in the aviation domain.
- **BANKING** dataset comprises 13,083 utterances related to the banking domain, encompassing 77 different fine-grained intents. It serves as a valuable resource for training and evaluating natural language processing models in tasks associated with banking-related user queries.
- **CLINC** dataset spans 10 domains, including travel, kitchen, utility, and more, encompassing a diverse set of 150 different intent classes. Designed for evaluating natural language understanding systems, CLINC provides a rich collection of user queries across various domains, making it a comprehensive resource for intent detection and classification tasks.
- **DSTC** dataset consists of slot annotations across four domains: buses, events, homes, and rental cars. Designed for dialogue state tracking, this dataset provides valuable information for developing and evaluating natural language processing systems in the context of multi-domain dialogue understanding.
- **HWU** dataset includes 64 intents spanning 21 domains, such as alarm, music, IoT, news, and calendar. Covering a wide range of user intents related to home activities, HWU serves as a comprehensive resource for training and evaluating natural language processing models in the domain of home workouts and smart home interactions.

---
[*]Corresponding Author
[*]Corresponding Author

- **MIT_RESTAURANT** [1] dataset is a semantically tagged training and test corpus presented in BIO format. Focused on restaurant-related queries, it provides annotated data designed to aid in natural language understanding tasks, specifically those involving restaurant information and interactions.
- **MIT_MOVIE** [1] comprises a semantically annotated training and test corpus in BIO format. Our implementation focuses on the "eng" corpus, featuring straightforward queries for enhanced simplicity and clarity.
- **TOP** consists of 44,000 utterances, each annotated with a hierarchical semantic representation.
- **SNIPS** consists of crowdsourced queries distributed among seven user intents, representing various levels of complexity.

# C  DATASET ORDER

Table 10 lists six random permutations of tasks for Intent Detection and Slot Filling in the experiments.

Table 10: Six random permutations of tasks for Intent Detection and Slot Filling

| Task | Order | Datasets |
|---|---|---|
| Intent Detection | Order 1 | TOP_S1, HWU, SNIPS, BANKING, CLINC, TOP_S2, TOP_S3, ATIS |
| | Order 2 | BANKING, HWU, TOP_S1, TOP_S3, CLINC, TOP_S2, SNIPS, ATIS |
| | Order 3 | SNIPS, ATIS, TOP_S2, TOP_S3, CLINC, BANKING, HWU, TOP_S1 |
| | Order 4 | CLINC, SNIPS, TOP_S3, BANKING, TOP_S2, HWU, TOP_S1, ATIS |
| | Order 5 | BANKING, TOP_S2, TOP_S1, ATIS, TOP_S3, HWU, CLINC, SNIPS |
| | Order 6 | CLINC, TOP_S1, TOP_S2, ATIS, SNIPS, HWU, BANKING, TOP_S3 |
| Slot Filling | Order 1 | MIT_MOVIE, DSTC, MIT_RESTAURANT, SNIPS, ATIS |
| | Order 2 | MIT_MOVIE, SNIPS, DSTC, MIT_RESTAURANT, ATIS |
| | Order 3 | ATIS, MIT_MOVIE, DSTC, MIT_RESTAURANT, SNIPS |
| | Order 4 | DSTC, MIT_RESTAURANT, MIT_MOVIE, ATIS, SNIPS |
| | Order 5 | MIT_MOVIE, ATIS, SNIPS, MIT_RESTAURANT, DSTC |
| | Order 6 | SNIPS, ATIS, MIT_RESTAURANT, MIT_MOVIE, DSTC |

---

[1] https://groups.csail.mit.edu/sls/downloads/