# OpenReview forum: "Dirichlet Continual Learning: Tackling Catastrophic Forgetting in NLP"
_auai.org/UAI/2024/Conference — UAI 2024 poster_

### Official Review · Reviewer_4dqP · 2024-03-22

**Q2-1 Originality-Novelty:** 3
**Q2-2 Correctness-Technical Quality:** 3
**Q2-5 Clarity Of Writing:** 3

**Q1 Summary And Contributions:**

This work deals with the problem of continual learning, which employs neural networks which suffer from catastrophic forgetting.  To avoid that behaviour, a method called generative-based rehearsal is employed as a means to reduce the cost of retraining, but it may fail to accurately capture the task-specific distribution. A special form of generative-based rehearsal called Dirichlet Continual Learning is proposed to capture the task-specific distributionof latent variables.  Furthermores, it intorduces a new distillation method to be used during sample generation.  Experiments are shown for task-oriented dialogue-systems.

**Q2-3 Extent To Which Claims Are Supported By Evidence:**

3: Good: the main claims are supported by convincing evidence (in the form of adequate experimental evaluation, proofs, (pseudo-)code, references, assumptions).

**Q2-4 Reproducibility:**

3: Good: key resources (e.g. proofs, code, data) are available and key details (e.g. proofs, experimental setup) are sufficiently well-described for competent researchers to confidently reproduce the main results.

**Q3 Main Strengths:**

The paper addresses a hard problem in continual learning, avoiding catastrophic forgetting via a well crafted method for pseudo samples generation that obeys a downstream application. Theoretical developments are provided to support the method. The method is tested for continual learning in dialogue system , with good results and comprehensive testing.

**Q4 Main Weakness:**

As Continual Learning is sensitive to downstream application, your claims should have been restricted to continual learning dialogue systems.  It is also not clear if Dirichlet Continual Learning  consists of the application of Jensen-Shannon Knowledge Distillation to generate  pseudo examples, or if those are two distinct methods that interact.

**Q5 Detailed Comments To The Authors:**

Please clarify if Dirichlet Continual Learning  consists of the application of Jensen-Shannon Knowledge Distillation to generate  pseudo examples, or if those are two distinct methods that interact.  If the latter is the case, please comment on the kind of interaction.  Why are ablation tests a comparison of DCL with PCLL?  It seems that ablation tests should contemplate  DCL without Jensen-Shannon Knowledge Distillation, or some form of adding Jensen-Shannon Knowledge Distillation to a non-DCL method (if that makes sense).

**Q9 Complying With Reviewing Instructions:**

Yes

---

> ### Author Rebuttal · Authors · 2024-04-06
>
> We extend our sincere appreciation for your insightful suggestions and comments provided. Subsequently, we will provide a more detailed analysis of the issues raised.
>
> Q1: As Continual Learning is sensitive to downstream application, your claims should have been restricted to continual learning dialogue systems.
>
> A1: We conduct experiments on two tasks in dialogue systems for the reason of the fair comparison.  We argue that our DCL is applicable to general NLP tasks and the experimental results on Llama2 in Table 7 support that our DCL is scalable and can be generalized to general NLP tasks.
>
> Q2: It is also not clear if Dirichlet Continual Learning consists of the application of Jensen-Shannon Knowledge Distillation to generate pseudo examples, or if those are two distinct methods that interact.
>
> A2. Our DCL consists of two main modules in consecutive stages: the Dirichlet-guided Pseudo Rehearsal Module for pseudo samples generation and the Jensen-Shannon Knowledge Distillation (JSKD) model for robust training.
>
> It is noted that we apply only one model as stated in the last sentence of Sec. 3.2, “the LM shares parameters with the decoder of CVAE, enabling the construction of a **unified** model to address the CL problem”.  Hence, upon encountering the first task, there is no knowledge distillation, we employ Eq. (3) to train the parameters of the model on the task-1’s samples.  Subsequently, while training task-n (n≥2), we generate pseudo samples from the previously trained model first and we apply Eq. (3) to train the parameters of the current model on the task-n’s samples while updating the parameters of the current model by Eq. (10) on the generated pseudo samples. This iterative process allows the current model does not forget the knowledge in previous tasks.
>
> We have reported the ablation study results of these two modules in Table 2 and Table 3, respectively, to demonstrate their efficacy.
>
> Q3: Why are ablation tests a comparison of DCL with PCLL? It seems that ablation tests should contemplate DCL without Jensen-Shannon Knowledge Distillation, or some form of adding Jensen-Shannon Knowledge Distillation to a non-DCL method (if that makes sense).
>
> A3: The results in Table 2 are exclusively for the **Dirichlet-guided Rehearsal Module,** and Table 3 solely for the **JSKD approach**.  Table 2 presents the comparisons of effect of different latent variables on DCL and PCLL while Table 3 presents the effect of facilitating different knowledge distillation strategy on DCL, i.e., using only the Dirichlet-guided Rehearsal Module.
>
> Specifically, as stated in Sec. 4.6, under the paragraph of “**Gaussian vs. Dirichlet-guided Rehearsal Module”**, “We conduct a comparative analysis of DCL and PCLL in Intent Detection and Slot Filling to evaluate the impact of a Dirichlet-guided rehearsal module... The results presented in Table 2 demonstrate that DCL, with the Dirichlet-guided rehearsal module, consistently outperforms PCLL, which utilizes a Gaussian-guided module, across all evaluation metrics in both tasks…”
>
> As stated in the paragraph of “**KL Knowledge Distillation vs. JSKD**” in Sec. 4.6, “We evaluate the impact of JSKD on the performance of DCL and compare it with DCL equipped with KL Knowledge Distillation. …as summarized in Table 3, … Notably, we observe significant improvements ranging from 0.63% to 1.55% in the F1 score and 0.41% to 1.11% in LCA. This conclusion also holds for Intent Detection”.

---

### Official Review · Reviewer_3Cmz · 2024-03-23

**Q2-1 Originality-Novelty:** 1
**Q2-2 Correctness-Technical Quality:** 3
**Q2-5 Clarity Of Writing:** 2

**Q10 Ethical Concerns:**

No.

**Q1 Summary And Contributions:**

The authors propose Dirichlet continual learning as a novel generative-based rehearsal strategy for continual learning tasks in natural language processing. Further, they introduce a Jensen-Shannon knowledge distillation strategy to improve the pseudo-sample generation process. Experiments demonstrate that DCL outperforms state-of-the-art alternatives in two typical tasks (intent detection and slot filling) of task-oriented dialogue systems.

**Q2-3 Extent To Which Claims Are Supported By Evidence:**

3: Good: the main claims are supported by convincing evidence (in the form of adequate experimental evaluation, proofs, (pseudo-)code, references, assumptions).

**Q2-4 Reproducibility:**

2: Fair: key resources (e.g. proofs, code, data) are unavailable but key details (e.g. proof sketches, experimental setup) are sufficiently well-described for an expert to confidently reproduce the main results.

**Q3 Main Strengths:**

The authors present a relatively simple extension to PCLL by i) replacing the prior of the latent variable space from Gaussian to Dirichlet, and ii) by leveraging JSD-based knowledge distillation. Experimental results are somewhat convincing, specially the results in Table 3 supported by statistical analysis and the attention put into the ablation studies.

**Q4 Main Weakness:**

The methodological innovation is limited, however, it is not necessarily a weakness. Note though that the authors do not offer a compelling justification of why the Dirichlet prior is a better alternative than the more commonly used Gaussian prior. The experiments are more problematic, namely, details of hyperparameter optimization of the proposed model and baselines (especially PCLL) are not discussed, proper statistical analyses of the performance differences are only presented for Table 3, results in Table 3 are only presented for a single setting, results in Table 4 are presented for a single ratio for PCLL and results for DCL with KL are presented for a single value of z.

**Q5 Detailed Comments To The Authors:**

The last two big paragraphs of the introduction are quite repetitive. The space could be more efficiently used elsewhere.

Though the comprehensive related work section is welcome, some of its contents (not related to rehearsal) are less relevant to the proposed approach, in which case they may be safely removed to relegated to the Appendix.

Why is there a need to maximize both p_theta(g,y) and p_theta(y|g) (the joint and the conditional), when arguably p_theta(g,y) = p_theta(y|g)p(g) and g is just an empirical (point mass) distribution?

Since the results in Table 1 are averages over multiple runs, it is important to show the variation either in Table 1 or the supplement to better understand the differences between DCL, PCLL (the strongest baseline) and Multi. It is also important to be cautious with the language, because in the main results, an F1 difference of 2.89% in favor of the proposed model is notable whereas a difference of 3.43% in favor of Multi is only a "slight lag".

Though space limitations are argued for not presenting results for KL vs. JS in the intent detection experiment, is not clear why results are not simply presented in the Appendix. Also, not clear is the need to present results for all the orders as opposed to present average results with variation in both settings and reserve the detailed order results to the Appendix.

The statistical analysis for the results in Table 3 is a welcome addition, however it is not clear why a similar analysis was not carried out for results in Tables 1, 2 and the not-shown results for intent detection.

The reasoning behind changing the pseudo-sampling ratio in DCL but not in PCLL is not clear.

**Q9 Complying With Reviewing Instructions:**

Yes

---

> ### Author Rebuttal · Authors · 2024-04-06
>
> We extend our sincere appreciation for the suggestions and comments provided.
>
> Q1: the authors do not offer a compelling justification of why the Dirichlet prior is a better alternative than the more commonly used Gaussian prior.
>
> A1: As mentioned in Sec. 3.3, under the paragraph of “**Dirichlet-guided Pseudo Rehearsal Module**”, we mentioned “Dirichlet distribution in a discrete space, with α as the parameter vector. By leveraging the versatile mathematical properties of the Dirichlet distribution, our DCL model better approximates the actual token distribution, which can exhibit concave, convex, symmetrical, or asymmetrical behavior”.
>
> The Dirichlet latent distribution is considered superior to the Gaussian distribution due to its greater flexibility in mathematical structure. Unlike the Gaussian distribution, which possesses a symmetric structure, the Dirichlet distribution offers a more versatile representation. This flexibility enables the Dirichlet distribution to effectively model complex latent distributions that originate from discrete spaces, a capability that the Gaussian distribution lacks.
>
> The advantage of applying the Dirichlet latent distribution is also supported in the paragraph of “**Evaluating Pseudo Samples Quality**” of Sec. 4.6, “Table 5 summarizes the Dist-n results. Notably, DCL achieves higher distinct scores compared to other methods, indicating that DCL-generated pseudo samples exhibit larger diversity. This suggests that pseudo samples created by DCL are more similar to real samples.”
>
> Experiments show introducing Dirichlet latent variables can lead to significant improvement.
>
>
>
> Q2: details of hyperparameter optimization of the proposed model and baselines (especially PCLL) are not discussed
>
> A2: As PCLL is the SOTA baseline, to make a fair comparison, we have follow its setting and apply the same hyperparameters, the same datasets, dataset orders, latent variable dimension, epoch, learning rate, etc., as reported in PCLL.  It is noted that results about PCLL are obtained from [Zhao et al., 2022a], as explicitly stated in the caption of Table 1, “Scores of baselines are reported in [Zhao et al., 2022a]”.
>
>
> Q3: proper statistical analyses of the performance differences are only presented for Table 3, results in Table 3 are only presented for a single setting.
>
> A3: It is noted that results in Table 2 have demonstrated the effectiveness of applying the Dirichlet latent variables.  Subsequently, results in Table 3 show that the proposed JSKD strategy beats KLKD.  To provide more detailed results, we have listed the results in six order in Table 3.  As explicitly mentioned in the paragraph of “**KL Knowledge Distillation vs. JSKD.**” in Sec. 4.6, “Due to space limitations, we only report the performance in Slot Filling under different learning orders, as summarized in Table 3, where the performance in Intent Detection follows similar trends.”
>
> To meet the reviewer’s requirement, we provide the results of **Intent Detection** as follows.
>
> | Orders | Score[DCL with KL] | LCA[DCL with KL] | Score[DCL with JS] | LCA[DCL with JS] |
> | --- | --- | --- | --- | --- |
> | order 0 | 92.76 | 91.27 | 93.52 | 92.69 |
> | order 1 | 92.76 | 88.49 | 93.64 | 90.45 |
> | order 2 | 92.56 | 93.36 | 93.74 | 95.47 |
> | order 3 | 92.91 | 91.53 | 93.83 | 93.34 |
> | order 4 | 92.68 | 89.92 | 93.79 | 92.04 |
> | order 5 | 93.33 | 93.32 | 93.85 | 94.22 |
> | Mean | **92.83** | **91.32** | **93.73** | **93.04** |
>
>
>
> Q4: results in Table 4 are presented for a single ratio for PCLL and results for DCL with KL are presented for a single value of z.
>
> A4: As we apply the same hyperparameters of PCLL, we conduct experiments with varying ratios of pseudo samples in DCL to assess their impact.  Results in Table 4, i.e., in the paragraph of “**Number of Pseudo Samples**” in Sec. 4.6, show that DCL with a ratio of 0.1 outperforms PCLL with a ratio of 0.2, indicating even half the number of pseudo samples in DCL can lead to superior performance.  More results about different ratios of PCLL can be referred to [Zhao et al., 2022a].
>
> We also list the results of PCLL from [Zhao et al., 2022a] as follows
>
> | Ratio | Score | LCA |
> | --- | --- | --- |
> | 0.01 | 73.61 | 84.35 |
> | 0.05 | 84.09 | 89.54 |
> | 0.20 | 90.25 | 88.82 |
> | 0.50 | 91.02 | 91.44 |
> | 1.00 | **91.31** | **91.77** |
>
>
> Q5: The last two big paragraphs of the introduction are quite repetitive...may be safely removed to relegated to the Appendix.
>
> A5: Thank you so much for your advice, we will revise them in the final version.
>
>
> Q6: Why is there a need to maximize both p_theta(g,y) and p_theta(y|g) (the joint and the conditional)
>
> A6: The first term learns to decode the constructed sentence given the start token [BOS], and the second term learns to predict the output y after reading the prompt g(xi).
>
>
> Q7: It is also important to be cautious with the language...
>
> A7: Thanks for pointing out the wording issue.  We will remove the adjective words if needed.

---

### Official Review · Reviewer_9kxB · 2024-03-23

**Q2-1 Originality-Novelty:** 2
**Q2-2 Correctness-Technical Quality:** 3
**Q2-5 Clarity Of Writing:** 3

**Q1 Summary And Contributions:**

Summary: This paper addresses the challenge of continual learning in NLP by reducing of catastrophic forgetting. The approach focuses on using a generative-based rehearsal technique, generating pseudo-samples that mimic real data, thus enabling the model to preserve knowledge from previous tasks. The generative model uses Conditional Variational Autoencoders and incorporates Dirichlet distribution to model the latent prior variable, instead of the Gaussian distribution used in related works. Additionally, the authors introduce Jensen-Shannon Knowledge Distillation (JSKD) to enhance knowledge transfer during the generation of pseudo-samples.

Contributions: The authors suggest integrating Conditional Variational Autoencoders and Dirichlet latent variables, along with Jensen-Shannon Knowledge Distillation. Additionally, various experiments are conducted, including comparisons with baselines and conducting ablation studies.

**Q2-3 Extent To Which Claims Are Supported By Evidence:**

3: Good: the main claims are supported by convincing evidence (in the form of adequate experimental evaluation, proofs, (pseudo-)code, references, assumptions).

**Q2-4 Reproducibility:**

2: Fair: key resources (e.g. proofs, code, data) are unavailable but key details (e.g. proof sketches, experimental setup) are sufficiently well-described for an expert to confidently reproduce the main results.

**Q3 Main Strengths:**

The paper effectively addresses the challenge of continual learning in NLP and provides thorough research context, along with experiments demonstrating improvements over previous methodologies. The clarity of writing facilitates easy comprehension and follow-through for reviewers.

**Q4 Main Weakness:**

One main weakness of the paper is its novelty. While the main solution shares similarities with previous work like PCLL, the authors introduce advanced improvements. However, the similarity to previous works make this paper be incremental. Furthermore, the experiments primarily concentrate on two tasks within task-oriented dialogue, which might limit the ability to demonstrate the model's generalization across a broader range of NLP tasks.

**Q5 Detailed Comments To The Authors:**

It would be beneficial for the authors to incorporate comparisons with prompt-based approaches that have sota performance in the context of continual learning, following rehearsal-based, regularization, and architecture-based methods. By including such comparisons, the paper could provide a more comprehensive evaluation of the proposed approach and its effectiveness relative to other state-of-the-art techniques.

**Q9 Complying With Reviewing Instructions:**

Yes

---

> ### Author Rebuttal · Authors · 2024-04-06
>
> We express our sincere gratitude for the time and your valuable suggestions to review this paper. We will elaborate on the issue in the following.
>
> Q1: One main weakness of the paper is its novelty. While the main solution shares similarities with previous work like PCLL, the authors introduce advanced improvements. However, the similarity to previous works make this paper be incremental. Furthermore, the experiments primarily concentrate on two tasks within task-oriented dialogue, which might limit the ability to demonstrate the model's generalization across a broader range of NLP tasks.
>
> A1: DCL introduces the **Dirichlet-guided Pseudo Rehearsal** and **Jensen-Shannon Knowledge Distillation(JSKD)** in the context of continual learning for NLP. We argue that the primary contributions of DCL are its pioneering utilization of both methods in continual learning for NLP. Furthermore, extensive experimentation substantiates the superiority of DCL over baseline approaches.  More importantly, the experiments on Llama2 demonstrates its applicable to general NLP tasks.

---

### Official Review · Reviewer_MEBJ · 2024-03-23

**Q2-1 Originality-Novelty:** 2
**Q2-2 Correctness-Technical Quality:** 3
**Q2-5 Clarity Of Writing:** 2

**Q10 Ethical Concerns:**

No.

**Q1 Summary And Contributions:**

The paper introduces a new generative-based rehearsal strategy aimed at addressing the critical issue of catastrophic forgetting in continual learning. The authors suggest incorporating 1. a Dirichlet latent variable within the Conditional Variational Autoencoder (CVAE) framework and 2. employing Joint Stochastic Knowledge Distillation (JSKD) over traditional Knowledge Distillation (KD) through KL Divergence. This approach enhances the generation of pseudo samples. Empirical experiments demonstrate that the proposed method, DCL, achieves the upper bound and outperforms the performance of the current state-of-the-art generative-based rehearsal method, PCLL, in two standard tasks.

**Q2-3 Extent To Which Claims Are Supported By Evidence:**

3: Good: the main claims are supported by convincing evidence (in the form of adequate experimental evaluation, proofs, (pseudo-)code, references, assumptions).

**Q2-4 Reproducibility:**

2: Fair: key resources (e.g. proofs, code, data) are unavailable but key details (e.g. proof sketches, experimental setup) are sufficiently well-described for an expert to confidently reproduce the main results.

**Q3 Main Strengths:**

1. The proposal to replace the symmetric multivariate Gaussian distribution with a Dirichlet distribution is a novel approach.
2.  Empirical experiments demonstrate the effective of the DCL over the state-of-the-art state-of-the-art generative-based rehearsal method, PCLL.

**Q4 Main Weakness:**

- Is using Knowledge Distillation via Jensen-Shannon (JS) Divergence a new idea? I believe these works (https://arxiv.org/pdf/2110.15094.pdf, https://arxiv.org/pdf/2307.15190.pdf ) have explored the use of KD via JS but I do not see any citations in the paper.
- I'm interested in seeing a comparison of the running times between PCLL and DCL. A inclusion of Score vs. wall-clock time will be helpful.
- In reviewing section 4.6's ablation study, I'm unclear on the rationale behind comparing the integration of KL knowledge distillation in DCL to PCLL, and whether this constitutes a fair comparison. Could you provide further explanation?
- It would be beneficial if you could share the outcomes of experiments conducted exclusively with the JSDL approach and those using only the Dirichlet-guided Rehearsal Module. This information would help in understanding the individual contributions of each component.
- Given the introduction of Knowledge Distillation, there may be concerns regarding memory consumption in comparison to PCLL. Could you provide details on the memory usage? Specifically, is there a notable difference in memory consumption between the two approaches?

**Q5 Detailed Comments To The Authors:**

See main weakness for questions and improvement.

**Q9 Complying With Reviewing Instructions:**

Yes

---

> ### Author Rebuttal · Authors · 2024-04-06
>
> Q1: Is using Knowledge Distillation via Jensen-Shannon (JS) Divergence a new idea?
>
> A1: As stated in Sec. 3.4, we have cited the pioneer work of KD in CL, [Chuang et al., 2020].  We argue that applying Jensen-Shannon Knowledge Distillation (JSKD) in continual learning is novel and we have conducted sufficient experiment comparison in Sec. 4.6.  Although the reviewer has provided two referenced work, they are not related to continual learning.  We will elaborate them in the final version.
>
> Q2: I'm interested in seeing a comparison of the running times between PCLL and DCL.
>
> A2:  As indicated in Sec. 4.3, we highlight that "the training time for DCL on Intent Detection using a single GPU is approximately 5 hours, while for Slot Filling, it is around 6 hours." Additionally, we have internally conducted experiments on PCLL and observed that the running time of PCLL for Intent Detection is approximately 4.7 hours, and around 5.8 hours for Slot Filling. It is worth mentioning that DCL with KL knowledge distillation, the corresponding training time of DCL is 4.8 hours for Intent Detection and 5.9 hours for Slot Filling, respectively. We argue that the slight increase in training time is justified by the significant performance gain achieved.  In real-world deployment, the inference time of our DCL remains comparable with PCLL.
>
> The results from Tables 1 and 2 in the submitted manuscript, along with corresponding times, are provided again below for easy reference.
>
> | Model | Score(Intent) | LCA(Intent) | Score(Slot) | Score(Slot) | Time(Intent) | Time(Slot) |
> | --- | --- | --- | --- | --- | --- | --- |
> | PCLL | 90.25 | 88.82 | 74.48 | 68.41 | 4.7 hours | 5.8 hours |
> | DCL(with KL) | 92.83 | 91.32 | 76.42 | 73.76 | 4.8 hours | 5.9 hours |
> | DCL(with JS) | 93.73 | 93.04 | 77.37 | 74.49 | 5 hours | 6 hours |
>
> Q3: I'm unclear on the rationale behind comparing the integration of KL knowledge distillation in DCL to PCLL, and whether this constitutes a fair comparison. Could you provide further explanation?
>
> A3:  It is noted that different from PCLL, our DCL consists of two main modifications, applying Dirichlet latent variables in the Dirichlet-guided Rehearsal Module for pseudo samples generation and Jensen-Shannon Knowledge Distillation (JSKD) for robust model training.
>
> As in Sec. 4.6, Table 2 presents the comparisons of effect of latent variables, i.e., DCL with Dirichlet prior and PCLL with Gaussian prior, by setting the same knowledge distillation strategy  on them.  “…The results presented in Table 2 demonstrate that DCL, with the Dirichlet-guided rehearsal module, consistently outperforms PCLL, which utilizes a Gaussian-guided module, across all evaluation metrics in both tasks…”
>
> Moreover, Table 3 presents the effect of facilitating different knowledge distillation strategy on our DCL.
>
> Q4: It would be beneficial if you could share the outcomes of experiments conducted exclusively with the JSDL approach and those using only the Dirichlet-guided Rehearsal Module.
>
> A4: Indeed, we have reported the results in Table 2 exclusively for the **Dirichlet-guided Rehearsal Module,** and Table 3 solely for the **JSKD approach**.  Table 2 presents the comparisons of effect of different latent variables on DCL and PCLL while Table 3 presents the effect of facilitating different knowledge distillation strategy on DCL, i.e., using only the Dirichlet-guided Rehearsal Module.
>
> Specifically, as stated in Sec. 4.6, under the paragraph of “**Gaussian vs. Dirichlet-guided Rehearsal Module”**, “We conduct a comparative analysis of DCL and PCLL in Intent Detection and Slot Filling to evaluate the impact of a Dirichlet-guided rehearsal module... The results presented in Table 2 demonstrate that DCL, with the Dirichlet-guided rehearsal module, consistently outperforms PCLL, which utilizes a Gaussian-guided module, across all evaluation metrics in both tasks…”
>
> As stated in the paragraph of “**KL Knowledge Distillation vs. JSKD**” in Sec. 4.6, “We evaluate the impact of JSKD on the performance of DCL and compare it with DCL equipped with KL Knowledge Distillation. …as summarized in Table 3, … Notably, we observe significant improvements ranging from 0.63% to 1.55% in the F1 score and 0.41% to 1.11% in LCA. This conclusion also holds for Intent Detection”.
> Q5: Given the introduction of Knowledge Distillation, there may be concerns regarding memory consumption in comparison to PCLL. Could you provide details on the memory usage?
>
> A5: The memory costs of DCL and PCLL in Intent Detection are 34G and 27G, respectively, when the ratio of pseudo samples is 0.2 and the batch size is 32.  Moreover, the memory costs of DCL and PCLL in Slot Filling are 30G and 25G, respectively, when the ratio of pseudo samples is set to 0.2 and the batch size is 32.
>
> We argue that the increase in memory is justified by the significant performance gain achieved.  In real-world deployment, the inference time of our DCL remains comparable with PCLL.

---

### Official Review · Reviewer_CmRt · 2024-04-01

**Q2-1 Originality-Novelty:** 2
**Q2-2 Correctness-Technical Quality:** 3
**Q2-5 Clarity Of Writing:** 3

**Q1 Summary And Contributions:**

This work proposes a generative rehearsal based method for continual learning for NLP. More specifically they propose two changes to an existing method: using a Dirichlet latent prior for the conditional VAE and using JS divergence (symmetric KL) for KD. With these changes they demonstrate improved performance compared to existing methods  on intent detection and slot-filling.  Further experiments demonstrate the benefit of using JS divergence KD over the standard version.

**Q2-3 Extent To Which Claims Are Supported By Evidence:**

3: Good: the main claims are supported by convincing evidence (in the form of adequate experimental evaluation, proofs, (pseudo-)code, references, assumptions).

**Q2-4 Reproducibility:**

2: Fair: key resources (e.g. proofs, code, data) are unavailable but key details (e.g. proof sketches, experimental setup) are sufficiently well-described for an expert to confidently reproduce the main results.

**Q3 Main Strengths:**

1) Continual learning is an important problem for NLP specially for large language models.

2) Strong results on the intent detection and slot filling tasks.

**Q4 Main Weakness:**

1) The method makes two changes to an existing method. I am not sure these changes themselves are significant without strong empirical support.

2) Even though the results are good, experiments on Intent Detection and Slot filling only are not enough to make a general case for the efficacy of this method for continual learning.

**Q5 Detailed Comments To The Authors:**

1) Using the JS divergence for KD is not something new. It is a detail which KD papers have mentioned in passing in the experimental section. There has been a case for using forward KL, backward KL and symmetric KL (JS) in different KD papers. This is not something new, nor are the authors the first to use it for KD.

2) Whats the storage difference between saving a replay buffer for methods such as Adapter CL and saving a generative model and a KD model? What is the impact on training time ?

**Q9 Complying With Reviewing Instructions:**

Yes

---

> ### Author Rebuttal · Authors · 2024-04-06
>
> We sincerely thank you for the constructive suggestions and comments.  We will elaborate on the issues in the following.
>
> Q1: The method makes two changes to an existing method. I am not sure these changes themselves are significant without strong empirical support.
>
> A1: In Sec. 4.6, we have provided ablation study on “**Gaussian vs. Dirichlet-guided Rehearsal Module”** and “**KL Knowledge Distillation vs. JSKD**”.  The results in Table 2 and Table 3 have presented the significant contribution of Dirichlet-guided rehearsal module and JSKD, respectively.  We highlight the statements as follow:
>
> As in “**Gaussian vs. Dirichlet-guided Rehearsal Module**“, “We conduct a comparative analysis of DCL and PCLL in Intent Detection and Slot Filling to evaluate the impact of a Dirichlet-guided rehearsal module... The results presented in Table 2 demonstrate that DCL, with the Dirichlet-guided rehearsal module, consistently outperforms PCLL, which utilizes a Gaussian-guided module, across all evaluation metrics in both tasks…”
>
> As stated in “**KL Knowledge Distillation vs. JSKD**”, “We evaluate the impact of JSKD on the performance of DCL and compare it with DCL equipped with KL Knowledge Distillation. …as summarized in Table 3, … Notably, we observe significant improvements ranging from 0.63% to 1.55% in the F1 score and 0.41% to 1.11% in LCA. This conclusion also holds for Intent Detection”.
>
> Q2: Even though the results are good, experiments on Intent Detection and Slot filling only are not enough to make a general case for the efficacy of this method for continual learning.
>
> A2: We address the efficacy of our DCL in continual learning to general cases by three points:
>
> Firstly, in continual learning of NLP, most studies focus on only one or two tasks in natural language understanding, natural language generation, dialogue state tracking, or text classification. For instance, CPT[1] addresses dialogue state tracking, while ACM[2] tackles natural language generation. L2KD[3] tries to solve natural language generation and text classification tasks. To ensure fairness, we select two tasks, the exact ones in PCLL[4], which is the SOTA baseline, to demonstrate our DCL's efficiency.
>
> Secondly, most of existing works choose tasks in dialogue systems[5] due to their complexity.  Especially, in these tasks, generating sequential text across diverse domains remains a significant challenge.
>
> Lastly and more importantly, we have conducted experiments in Llama2, which contains generalization ability to general NLP tasks.  Due to the scaling law[6,7], we know that the pre-trained large language models (LLMs) are consistent. The excel performance of DCL on Llama2 has demonstrated our DCL is scalable and can be generalized to general NLP tasks.
>
> Q3: Using the JS divergence for KD is not something new. It is a detail which KD papers have mentioned in passing in the experimental section. There has been a case for using forward KL, backward KL and symmetric KL (JS) in different KD papers. This is not something new, nor are the authors the first to use it for KD.
>
> A3: DCL introduces the **Dirichlet-guided Pseudo Rehearsal** and **Jensen-Shannon Knowledge Distillation(JSKD)** in the context of continual learning for NLP. We argue that the primary contributions of DCL are its pioneering utilization of both methods in continual learning for NLP. Furthermore, extensive experimentation substantiates the superiority of DCL over baseline approaches.  More importantly, the experiments on Llama2 demonstrates its applicable to general NLP tasks.
>
> Q4: Whats the storage difference between saving a replay buffer for methods such as Adapter CL and saving a generative model and a KD model? What is the impact on training time ?
>
> A4: To elaborate, in both REPLAY and AdapterCL, the resource utilization scales linearly with the number of tasks. Specifically, in REPLAY, the episodic memory accumulates samples linearly, equivalent to 50 times the number of tasks. In AdapterCL, the parameter count increases linearly, corresponding to the number of adapter parameters multiplied by the number of tasks. In contrast, DCL employs a singular model to manage various tasks, as articulated in the concluding statement of Section 3.2: "the LM shares parameters with the decoder of CVAE, enabling the construction of a unified model to address the CL problem.”
>
> [1] Continual Prompt Tuning for Dialog State Tracking( Zhu et al., ACL2022)
>
> [2] Continual Sequence Generation with Adaptive Compositional Modules (Zhang et al., ACL2022)
>
> [3] Lifelong Language Knowledge Distillation. (Chuang et al., EMNLP2020)
>
> [4] Prompt Conditioned VAE: Enhancing Generative Replay for Lifelong Learning in Task-Oriented Dialogue (Zhao et al., EMNLP2022 )
>
> [5] Continual Learning in Task-Oriented Dialogue Systems. (Madotto et al., ACL2021)
>
> [6] Scaling Laws for Neural Language Models. (Kaplan et al., 2020)
>
> [7] Training Compute-Optimal Large Language Models. (Hoffmann et al., 2023)

---

### Meta-Review · Area_Chair_wj3U · 2024-04-22

After going over the paper, I am quite positive about it. The main contribution seems to be to make the standard derivation of EWC, i.e., the Laplace approximation, and replace the Gaussian prior with a Dirichlet prior. This makes sense in that the per-token distribution over the next word is a Categorical. All reviewers recommended acceptance -- one gave a strong accept. My endorsement of this paper is based not this as well as the fact that after skimming the paper, it looks like a paper I would really like to read in more depth. As the four reviewers found no technical flaws, I would be happy to see such a paper in the proceedings.